# Transcriptome-Wide Identification of CCCH-Type Zinc Finger Proteins Family in *Pinus massoniana* and RR-TZF Proteins in Stress Response

**DOI:** 10.3390/genes13091639

**Published:** 2022-09-13

**Authors:** Dengbao Wang, Sheng Yao, Romaric Hippolyte Agassin, Mengyang Zhang, Xuan Lou, Zichen Huang, Jinfeng Zhang, Kongshu Ji

**Affiliations:** Key Laboratory of Forestry Genetics & Biotechnology of Ministry of Education, Co-Innovation Center for Sustainable Forestry in Southern China, Nanjing Forestry University, Nanjing 210037, China

**Keywords:** CCCH zinc finger proteins, tandem CCCH zinc finger, *P. massoniana*, abiotic stresses, expression patterns

## Abstract

CCCH-type zinc finger proteins play an important role in multiple biotic and abiotic stresses. More and more reports about CCCH functions in plant development and stress responses have appeared over the past few years, focusing especially on tandem CCCH zinc finger proteins (TZFs). However, this has not been reported in Pinaceae. In this study, we identified 46 CCCH proteins, including 6 plant TZF members in *Pinus massoniana*, and performed bioinformatic analysis. According to RT-PCR analysis, we revealed the expression patterns of five RR-TZF genes under different abiotic stresses and hormone treatments. Meanwhile, tissue-specific expression analysis suggested that all genes were mainly expressed in needles. Additionally, RR-TZF genes showed transcriptional activation activity in yeast. The results in this study will be beneficial in improving the stress resistance of *P. massoniana* and facilitating further studies on the biological and molecular functions of CCCH zinc finger proteins.

## 1. Introduction

Plant transcription factors (TFs) are widely involved in regulating the development and growth of plants. In recent years, the increasing reports with regard to the identification of various kinds of TFs in plants have shown the significance of TFs. The progress in characterizing TFs reveals that most TFs are involved in the regulation of stress-related genes in plants [1]. For example, some large TF families, such as bZIP (basic leucine zipper) [2], WRKY [3], AP2/ERF [4], MYB [5], bHLH (basic helix-loop-helix) [6], NAC [7], HD-ZIP [8], etc., have been identified in the past few years and have shown that they can respond to various stresses. Zinc-finger transcription factors act as one of the largest transcription factor (TF) families in plants, which are critical regulators for multiple biological processes, such as morphogenesis, signal transduction and environmental stress responses. Most CCCH proteins can widely participate in plant response to abiotic stresses, such as drought, oxidative, salinity, flooding and cold temperature stresses, and may perform both transcriptional and post-transcriptional regulation [9].

CCCH-type zinc finger proteins are composed of three cysteines and one histidine. Compared to most zinc fingers, such as the C2H2-type or CCCC-type, there are few studies on the less-common CCCH zinc finger proteins, and they have not been reported in gymnosperms. So far, the CCCH family have been identified in many eukaryotes. For example: there are 68 and 67 CCCH members in *Arabidopsis thaliana* and *Oryza sativa* [10], 91 members in *Populus trichocarpa* [11], 68 members in *Zea mays* [12], 34 in *Medicago truncatula* [13], 36 in *Aegilops tauschii* [14], 80 in *Solanium lycopersicum* [15], 69 in *Vitis vinifera* [16], 58 in *Cicer arietinum* [17] and 103 in *Brassica rapa* ssp. *Pekinensis* [18].

Common CCCH-type zinc finger proteins possess 1–6 copies of CCCH zinc finger motifs, whose consensus sequence is defined as C-X4-15-C-X4-6-C-X3-H [19]. The TZF motifs in mammals consist of two identical C-x8-C-x5-C-x3-H zinc fingers separated by 18 amino acids [20]. It is also found in plants, such as *PtrC3H17*, *PtrC3H18* and *PtrC3H20* [11], as well as AtC3H14, AtC3H15, OsC3H9 and OsC3H39 [10], but this is less common. The common TZF motifs in plants have a plant unique arginine-rich (RR) region in front of the TZF motif: the number of amino acids between two separated CCCH motifs (C-X7-8-C-X5-C-X3-H and C-X5-C-X4-C-X3-H) is 16. The CCCH proteins with plant TZF motifs are called RR-TZFs. RR-TZFs have been more deeply identified and studied in many model plants, including *Arabidopsis*, rice and poplar. Meanwhile, zinc finger proteins, which have no tandem motifs, are called non-TZF proteins [21]. For the past few years, the molecular function about plant TZF CCCH zinc finger genes have been studied more in-depth than non-TZF genes.

Except for the typical CCCH structures, there are some other vital domains which may influence the biological functions of CCCH zinc finger proteins. In the existing studies, many identified that CCCH proteins contain a nuclear export signal sequence (NES) or a nuclear localization sequence (NLS), which determines their subcellular localization. Some CCCH proteins have both an NES and NLS called typical shuttling proteins, which may shuttle between the cytoplasm and the nucleus [22].

*P. massoniana* is widely distributed in southern China, whose production comprises important raw material for the chemical industry, such as turpentine and rosin. It has great economic and ecological benefits. Due to the degradation of the ecological environment caused by global warming, *P. massoniana* is faced with a variety of adverse environments, which affect the growth of *P. massoniana* and restrict the development of the related industry. To adapt to variable environmental conditions, plants have evolved diverse regulatory pathways to resist stress; therefore, it is necessary to identify related genes which can help *P. massoniana* to resist abiotic stress. The plant genomes encode large numbers of CCCH zinc finger proteins which can play key roles in regulating the tolerance of plants to abiotic stresses. Due to a lack of genome of *P. massoniana*, we used transcriptomes to identify the CCCH zinc finger proteins family as soon as possible. In this study, we identified 46 CCCH members in *P. massoniana* and analyzed them through bioinformatics approaches. Furthermore, we analyzed the expression of five selected RR-TZF protein genes under different abiotic stresses and in different tissues. Additionally, because CCCH zinc finger proteins can function as activators of transcription [23], we constructed recombinant vectors to verify their transcriptional activation activity in the yeast system. The results aim to provide a reference to understand the molecular regulation mechanism of the CCCH gene family under abiotic stress and promote further studies on TZF proteins.

## 2. Materials and Methods

### 2.1. Identification of CCCH Genes in *Pinus massoniana*

We acquired the hidden Markov model (HMM) profile of the CCCH domain (PF00642) from the Pfam database (http://pfam.xfam.org/, accessed on 16 April 2022) [24]. The HMM profile was used to search the CCCH members from the following three transcriptomes. Transcriptome data for *P. massoniana* were derived from the previously determined CO_2_ stress transcriptome [25], drought stress transcriptome (PRJNA595650) and young shoots transcriptome (PRJNA655997). A BLASTP search was performed against three transcriptome data using the hidden Markov model (HMM) profile. A transcription factor prediction tool (http://planttfdb.gao-lab.org/prediction.php, accessed on 16 April 2022) was used to predict CCCH proteins. Then, we used Pfam and NCBI’s Conserved Domain search (CD Search) (https://www.ncbi.nlm.nih.gov/Structure/cdd/wrpsb.cgi, accessed on 16 April 2022) to check the predicted CCCH domain of initial CCCH TFs. Finally, sequences with complete CCCH domains were selected after deleting sequences with more than 97% similarity. The molecular weights and isoelectric points of the identified *PmC3H* proteins were computed from the ExPASy (Swiss Institute of Bioinformatics) website (https://web.expasy.org/compute_pi/, accessed on 17 April 2022).

### 2.2. Bioinformatics Analysis of PmC3H Proteins

Phylogenetic trees of the amino acid sequences were constructed using the maximum likelihood (ML) method with MEGA-X software, using 1000 bootstrap replicates. The CCCH protein sequences of *A. thaliana* and *P. tabuliformis* were downloaded from supplemental information in related articles [10,26]. The phylogenetic tree was edited for visualization purposes with the online software EvolView (https://www.evolgenius.info/evolview/#login, accessed on 9 August 2022). In order to better determine the distribution of conserved motifs and groups of *PmC3H* proteins, the online program Multiple Expectation Maximization for Motif Elicitation (MEME) (http://meme-suite. org/tools/meme, accessed on 12 April 2022) was used to analyze the conserved motifs of *PmC3H* proteins, with the number of motifs set to 10. The multiple sequence alignment of several TZF CCCH proteins’ conserved region was conducted in SnapGene (https://www.snapgene.com/, accessed on 12 April 2022) using MUSCLE. The types of CCCH motifs were determined by a regular expression search of (C.{7,8}C.{4,5}C.{3}H) in Notepad++.

The subcellular localization of *PmC3H* proteins were predicted and analyzed with CELLO (http://cello.life.nctu.edu.tw/, accessed on 12 April 2022), WoLF PSORT (https://wolfpsort.hgc.jp/, accessed on 12 April 2022), and Plant-mPLoc (http://www.csbio.sjtu.edu.cn/bioinf/plant-multi/). The NLS was predicted by SeqNLS (http://mleg.cse.sc.edu/seqNLS/, accessed on 15 April 2022). NES used the developed and widely accepted NES consensus: [LV]-x(2,3)-[LIVFM]-x(2,3)-L-x-[LIMTKD] [27]. We used regular expression search of [LV].{2,3}[LIVFM].{2,3}L. [LIMTKD] in Notepad++ to see the NESs’ distribution in all *PmC3H* proteins.

### 2.3. Plant Materials and Abiotic Stress Treatments

Two-year-old *P. massoniana* seedlings were obtained from the Key Laboratory of Forestry Genetics & Biotechnology of Ministry of Education (Nanjing Forestry University). Eight representative tissues (shoot apices; young needle; old needle; young stem; old stem; xylem; phloem; and root) were collected from three seedlings with the same height in the same pot. The seedlings were treated with nine abiotic stresses. The drought treatment was performed following the method in which samples were collected five times across 20 days through natural evaporation after watering at day 0. The field capacity was as follows: 0 d (67%); 3 d (63%); 7 d (58%); 12 d (46%); 20 day (34%). The mechanical damage treatment method was performed by cutting the upper half of the needles. Osmotic stress was induced by soaking the plants in the 15% polyethylene glycol (PEG6000) solution and 200 mM NaCl solution. For plant hormones treatment, the selected seedlings were sprayed independently on the needles’ surface with the following: 100 µM ABA (abscisic acid); 1 mM SA (salicylic acid); 10 mM H_2_O_2_ (hydrogen peroxide); 10 mM MeJA (methyl jasmonate); and 50 µM ETH (ethephon) solution (50 mL) [3]. Afterwards, needles were sampled at 0 h, 3 h, 6 h, 12 h and 24 h after treatments, except drought stress. The needles under drought were sampled at 0 d, 3 d, 7 d, 12 d and 20 d. Untreated seedlings served as the controls. All treatments were conducted on three biological replicates and three technological replicates.

### 2.4. RNA-Seq Data Analysis

Information about the drought treatment of *P**. massoniana* for the Illumina RNA-seq is shown below. One-year-old *P. massoniana* seedlings were obtained from the *P. massoniana* National Forest Seed Base in Duyun city, Guizhou Province, China. They were transplanted in a greenhouse for 3 months of adaptive growth in the Nanjing Forest University. A total of 120 healthy *P. massoniana* seedlings with the same height were selected in preparation for subsequent experiments. Soil water content was controlled by a weighing method. We set four gradients: normal water treatment (CK), mild (T1), medium (T2), serious (T3). Each field capacity was set to: CK (80 ± 5)%, T1 (65 ± 5)%, T2 (50 ± 5)% and T3 (35 ± 5)%. Each gradient had 10 samples repeated three times. The indoor temperature was controlled at 15–22 °C and the humidity was about 75%. All seedlings were grown under 16 h light culture (35,000 lx light intensity) and 8 h dark culture. Drought stress lasted for 60 days. Fragments per kilobase of the exon model per million reads mapped (FPKM) values were calculated to estimate the abundance of *PmC3H* gene transcripts. TBtools (Toolbox for Biologists) [28] software was used to create heat maps of partial genes based on the values of log2 (FPKM + 0.01), and analyses were performed at the row scale. The color scale represents relative expression levels based on the values of the log2 fold change scale in different gradients.

### 2.5. RNA Extraction and qRT-PCR Analysis

Total RNA was extracted using the FastPure Plant Total RNA Isolation Kit (RC401, Vazyme Biotech, Nanjing, China). RNA concentration and purity were measured with a NanoDrop 2000 (Thermo Fisher Scientific, Waltham, MA, USA), and RNA integrity was estimated by 1.2% agarose gel electrophoresis. First-strand cDNA was synthesized using the One-step gDNA Removal and cDNA Synthesis Kit (AT311, TransGen Biotech, Beijing, China). Primers for quantitative real-time reverse transcription PCR (qRT-PCR) were designed using Primer 5.0 (Appendix A). SYBR Green reagents were used to detect the target sequence. Each PCR mixture (10 µL) contained 1 µL of diluted cDNA (20× dilution), 5 µL of SYBR Green Master Mix (11184ES03, Yeasen Biotech, Shanghai, China), 0.4 µL of each primer (10 µM), and 3.2 µL of ddH_2_O. The PCR program stages were as follows: (1) 95 °C for 2 mins (preincubation); (2) 95 °C for 10 s, (3) 60 °C for 30 s, repeated 40 times. The remaining steps used the instrument default settings. The PCR quality was estimated based on melting curves. The *α-tubulin* (*TUA*) gene was used as a reference gene [29]. Three independent biological replicates and three technical replicates for each biological replicate were examined. Quantification was achieved using comparative cycle threshold (Ct) values, and gene expression levels were calculated as 2(−∆∆Ct) [∆CT = CT Target−CT *TUA*. ∆∆Ct = ∆Ct Target−∆Ct CK]. A Duncan test was used to examine the significance between different columns in IBM SPSS Statistics (Version 25). The results were marked with lowercase letters, starting with the largest average. The same lowercase letters between different columns indicated no significant difference. Completely different lowercase letters between different columns indicated a significant difference, *p* < 0.05. More than one lowercase letter in the same column indicated no significant difference between the column and other columns which contained one of the lowercase letters. For example, a column marked “ab” meant there was no significant difference between the “ab” column and the “a/b” column.

### 2.6. Transcriptional-Activation Activity Assay

We cloned the coding regions of five genes (Appendix A) to investigate CCCH proteins’ transcriptional activation. The GAL4 DNA-binding domain of pGBKT7 vector was fused with the entire open reading frame (ORF) of *PmC3H20*, *PmC3H32*, *PmC3H40*, *PmC3H44* and *PmC3H45*, and the empty pGBKT7 vector was used as a negative control. The primers for constructing recombinant vector are listed in Appendix A. Then, we transformed these fusion vectors into yeast strain AH109 (YC1010, Weidi Biotechnology, Shanghai, China) according to the operational approach. The transformed yeast strains were screened on selective YPDA medium plates without tryptophan (SD/−Trp) and cultivated at 29 °C for 48 h. A single yeast strain was collected in 10 µL ddH_2_O and validated by PCR. After positive detection, the remaining positive bacterium liquid was diluted in 200 uL ddH_2_O, and 5 μL diluted fluid was collected and cultivated on the surface of different yeast mediums which, respectively, lacked tryptophan/histidine (SD/−Trp/−His) and tryptophan/histidine/leucine (SD/−Trp/−His/−Leu). The later medium had X-α-Gal on the surface. At the end, we took photographs to record the growth of yeast.

## 3. Results

### 3.1. Identification of CCCH Proteins in P. massoniana

We identified 121 putative CCCH genes from the transcriptome data. Afterwards, SMART and Pfam analyses were performed to verify the conserved domains. Finally, we obtained a total of 46 CCCH protein sequences. The 46 CCCH TFs were named from *PmC3H1* to *PmC3H46*. The coding sequences are available in Appendix A. Among the 46 CCCH proteins, the number of amino acids was between 126 and 956, with the predicted molecular weight ranging from 14.4 to 104.6 kDa. The pI values ranged from 5.48 to 10.8. The detailed range of protein molecular weights and the isoelectric point values are listed in Appendix A.

### 3.2. Phylogenetic Analysis

The ML method was used to construct a phylogenetic tree of 46 *PmC3H* protein sequences, as shown in Figure 1. The 46 CCCH protein sequences were divided into eight groups according to the phylogenetic tree, named from A to H. The group with the largest number of *PmC3H* proteins was G, with a total of 14 members, followed by F, which contained 8 members. The group with the smallest number was D, with only one member. In addition, a phylogenetic tree constructed by 46 *P. massoniana* CCCH protein sequences and 68 *Arabidopsis* CCCH protein sequences is provided in Appendix A; a phylogenetic tree constructed by 46 *PmC3Hs* and 49 *PtC3Hs* is shown in Appendix A. The whole group A containing six *PmC3H* members was completely classified with the largest subfamily IX (11 members) in *A. thaliana*. The whole group F of 8 *PmC3H* members was completely clustered with the largest subfamily I in *A. thaliana* (11 members). By further comparison, we found highly homologous genes with group A in *P. tabulaeformis* (Appendix A). By clustering analysis, we found seven RR-TZF genes (*PtC3H38*, *PtC3H30*, *PtC3H36*, *PtC3H51*, *PtC3H80*, *PtC3H81*, *PtC3H82*) in *P. tabulaeformis* (Appendix A). The homology of RR-TZFs between this two Pinaceae species ranged from 97.8% to 99.4%, indicating that RR-TZFs are highly conserved among two species. The protein sequences of 49 selected CCCH *P. tabulaeformis* protein sequences and 68 *A. thaliana* CCCH sequences are shown in Appendix A.

### 3.3. Protein Domain Analysis of PmC3H Proteins

According to the results of CELLO, Wolf PSORT and Plant-mPLoc, almost all CCCH genes’ subcellular localization were predicted to be at the nucleus. The results are listed in Appendix A. According to MEME program identification of the conserved motifs of the 46 *PmC3H* proteins, we found that all proteins had highly conserved CCCH domains. Ten conserved motifs are listed in Appendix A and illustrated in Figure 2. The amino acid length of the 10 motifs ranged from 15 to 50. Motif 1 was identified in all *PmC3H* genes, which is the basic structure of CCCH Motif 2, and Motif 3 was exclusively found in group A, which was essential to the RR-TZF conserved domain. Motif 4 was widely distributed in group F&H. However, almost all members in group F contained Motif 6, Motif 9 and Motif 10. Additionally, almost all group H members contained Motif 5. Motif 7 was exclusively found in group F. Motif 8 was exclusively distributed in group H. According to the phylogenetic tree, these conserved motifs in the same groups support the reliability of these group classifications.

The type and quantity of each CCCH motif were further determined by regular expression shown in Appendix A. We found a total of 119 CCCH motifs in all selected proteins and the most common motif in *P. massoniana* was C-X7-8-C-X5-C-X3-H, whose number reached 100, which accounted for 84% of all CCCH motifs. All *PmC3H* proteins with at least one common motif, except *PmC3H43*, only contained the motif: C-X9-C-X5-C-X3-H. In addition, we found that the majority of *P. massoniana* zinc finger proteins are composed of 1–6 CCCH copies. However, *PmC3H1* and *PmC3H10* have seven copies of CCCH motifs. Some unconventional CCCH motifs are as follows: C-X7-C-X6-C-X3-H (*PmC3H38* with 2 and *PmC3H39* with (1); C-X7-C-X4-C-X3-H (*PmC3H15*, *PmC3H25*, *PmC3H27*, *PmC3H42* with 1 and *PmC3H37* with (2); C-X11-C-X7-C-X4-H (*PmC3H5* with 1).

In addition to typical CCCH motifs, we also found that all members of group A contained a unique plant RR-TZF motif, C-X7-C-X5-C-X3-H-X16-C-X5-C-X4-C-X3-H, by multiple sequence alignment (Figure 3a). For those that had a representative structure (tandem CCCH motifs), they were preceded by an arginine-rich region. Additionally, *PmC3H16*, *PmC3H33* and *PmC3H35* contained a TZF motif with two identical C-X8-C-X5-C-X3-H zinc fingers in tandem, separated by 18 or 19 amino acids (Figure 3b). Furthermore, we found that the *PmC3H33* contained a EELR domain which was confirmed to be essential for the transcriptional activation [30].

In addition to the above CCCH structures, we found that some CCCH proteins in *P. massoniana* also had a leucine-rich NES and NLS. The SeqNLS prediction of 46 *PmC3H* proteins showed that the 18 *PmC3H* proteins with a predicted NLS had the NLS sequence with the highest score of matches within the predictions listed in Appendix A. We used a regular expression search of ([LV].{2,3} [LIVFM].{2,3}L. [LIMTKD]) in Notepad++ to see the distribution of NESs in all *PmC3H* proteins. Finally, 22 (47.8%) *PmC3H* proteins were found by nuclear export signal (NES) using the developed and widely accepted NES consensus [LV]-x(2,3)-[LIVFM]-x(2,3)-L-x-[LIMTKD] [30] (NESbase version 1.0: a database of nuclear export signals [J]. Nucleic Acids Research), and the results are listed in (Appendix A).

According to the results of phylogenetic analysis and multiple sequence alignment, we selected five members of group A proteins for further RT-PCR expression analysis: *PmC3H20*, *PmC3H32*, *PmC3H40*, *PmC3H44* and *PmC3H45*. Due to the fact that *PmC3H23* and *PmC3H45* have a similarity of more than 90.8% and have the same conserved domain, we only selected *PmC3H45* as a representative.

### 3.4. Analysis of the Transcriptional Profiles of PmC3H Genes

The heat maps of partial CCCH genes were drawn based on drought stress transcriptome data (Figure 4). The expression levels of some genes in the transcription group under drought stress were too low to be detected, so only 30 expressions of *PmC3H* genes were given. The heatmap showed the changes in the expression of partial genes under different field capacity levels. For example, 12 of these genes (*PmC3H21*, *PmC3H31*, *PmC3H2*, *PmC3H36*, *PmC3H30*, *PmC3H15*, *PmC3H4*, *PmC3H23*, *PmC3H7*, *PmC3H25*, *PmC3H9* and *PmC3H12*) tended to increase with mild drought treatment, then decrease under higher levels of drought; all of them showed lower expression compared to CK. Seven genes (*PmC3H11*, *PmC3H22*, *PmC3H43*, *PmC3H6*, *PmC3H8*, *PmC3H26* and *PmC3H34*) showed a similar trend to the previous twelve genes with mild and medium drought, but they increased again under the serious drought. Six genes (*PmC3H13*, *PmC3H38*, *PmC3H17*, *PmC3H37*, *PmC3H32*, *PmC3H10* and *PmC3H19*) decreased with mild drought, increased under medium drought and then decreased again under serious drought. All of them showed higher expression compared to CK, except *PmC3H32*. In addition, *PmC3H14* showed a continuous increase under drought stress and *PmC3H3* showed a continuous decrease. The remaining genes showed no similar changes in expression. In general, almost all expression levels of *PmC3H* genes rose under different levels of drought treatment, indicating that *PmC3H* genes could have specific functions in resistance to drought stress.

### 3.5. Expression Levels of PmTZF Genes in Different Tissues

The qRT-PCR results in Figure 5 show the expression pattern of five RR-TZF genes in eight different tissues: shoot apices (T); young needle (YN); old needle (ON); young stem (YS); old stem (OS); xylem (X); phloem (P); root (R). The results show that five genes were expressed in all organs. The expression of *PmC3H20* and *PmC3H45* in young needles was higher than other tissues, and the expression of *PmC3H40* and *PmC3H44* in old needles was the highest. The expression of *PmC3H20* in young needles was 1.9–4 times higher than other tissues. The expression of *PmC3H45* in young needles was 9.4 times higher than shoot apices and 1.83.4 times than other tissues. The expression of *PmC3H40* in old needles was significantly higher than young needles, 15.7 times higher than shoot apices and 2.1–8.1 times higher than other tissues. *PmC3H44* was expressed the highest in needles and *PmC3H32* was expressed more in roots than other tissues. Their expression levels in other tissues were similar. In general, all five genes were mainly expressed in needles in *P. massoniana*, and their expression was obviously low in shoot apices: there was no significant difference in stem, xylem and phloem.

### 3.6. Expression Levels of PmRR-TZF Genes under Abiotic Stress

The results in Figure 6 show the responses of five RR-TZF genes to different abiotic stresses. Under ABA stress (Figure 6a), although 4 genes increased slightly at 3h, all of them significantly decreased at 24 h. All genes were induced significantly by MeJA, especially *PmC3H45* (Figure 6b). After the mechanical injury (Figure 6c), *PmC3H32* and *PmC3H44* showed a positive response, while the remaining genes were not sensitive to mechanical injury. Under SA treatment (Figure 6d), *PmC3H20* and *PmC3H32* showed a steady decline, and the expression of *PmC3H40*, *PmC3H44* and *PmC3H45* increased first, reached the peak and then decreased. Under H_2_O_2_ treatment (Figure 6e), the expression of *PmC3H20*, *PmC3H32* and *PmC3H40* decreased first and increased after. *PmC3H44* was not sensitive to H_2_O_2_ treatment, while the expression of *PmC3H45* was showed three times more at 0 h. Four genes were induced significantly by ETH (Figure 6f), except *PmC3H20*, although all five genes were induced by NaCl (Figure 6g), and their expression changed slightly. Under PEG treatment (Figure 6h), only the expression of *PmC3H40* decreased, and the remaining four genes increased sharply, especially *PmC3H32* and *PmC3H44*, which increased 12.9 and 7.8 times more at 0 h, respectively. Under drought treatment (Figure 6i), the expression levels of the five genes changed slightly in the first few days and increased significantly when *P. massoniana* seedlings were under drought stress at the 20th day. A concise summary of the expression profiles of the five RR-TZF genes under abiotic stress is listed in Appendix A.

### 3.7. Transcriptional Activation Activity

The photographs of the yeast growth assay (Figure 7) show that all five RR-TZF genes can activate transcription in a yeast system. The results suggest that these RR-TZF genes can function as a transcriptional regulator in *P. massoniana*.

## 4. Discussion

Plant genomes encode more TZF genes than animals due to genome-wide dispersed duplication [19]. More and more studies have shown that plant CCCH zinc finger proteins can contribute to resisting abiotic stress; therefore, it is necessary to identify CCCH zinc finger proteins in *P. massoniana*. Since it is difficult to obtain the high-quality genome of *P. massoniana*, we used transcriptome data to identify 46 *PmC3H* proteins. However, we downloaded all *PtC3H* (83) genes from the supplemental information of the Chinese pine genome published this year [26]. After deleting the sequences with more than 97% identity and verifying the conserved domains with the above methods, 49 *PtC3Hs* were finally identified, three more than we identified in *P. massoniana*. In contrast, we identified *PmC3H* proteins as comprehensively as possible. By further comparison, we found that RR-TZF genes are highly conserved and homologous in two Pinaceae species. This indicated that RR-TZF genes are highly conserved during the evolution of Pinaceae. Given that there are no reports on the function of the CCCH genes in Pinaceae, this systematic research lays the foundation for further studies on CCCH genes in *P. massoniana*.

Similar to 69.2% of poplar CCCHs with at least two CCCH motifs [11], there are 33 (71.7%) *PmC3Hs* which also contained at least two CCCH motifs. The motif C-X7-8-C-X5-C-X3-H was found in almost all *PmC3Hs*, which may be considered an ancestor of other CCCH motifs. In addition to conventional and non-conventional CCCH motifs in *P. massoniana*, we found a unique C-X11-C-X7-C-X4-H in *PmC3H5*, which was not mentioned in other plants. Additionally, we also found this unique motif in *P. tabuliformis* (*PtC3H12*), suggesting that *PmC3H5* containing this motif may have different binding activities and biological functions in Pinaceae.

Proteins must be localized at the appropriate region to perform its function; that means subcellular localization can offer clues to study the function of a protein. In the current literature, CCCH zinc finger proteins show several subcellular localization patterns: some are located in the nucleus; some are located in the cytoplasm plasma membrane; some are located in the cytoplasm; and some can shuttle between the cytoplasm and the nucleus [9]. At first, *Arabidopsis* CCCH TFs were predicted to be located in the nucleus and verified, such as *AtSZF1* [31]. However, the subsequent NES prediction suggest that 79.4% *Arabidopsis* CCCH genes may be nucleocytoplasmic shuttle proteins [10]. Furthermore, 11 *Arabidopsis* RR-TZF genes containing the NES were confirmed to be able to shuttle between the nucleus and cytoplasm [32,33], co-localizing with processing bodies (PBs) and stress granules (SGs) in the cytoplasm. PBs and SGs play an important role in plant tolerance to abiotic stress [34]. Not surprisingly, all *P. massoniana* CCCH members were predicted to localize in nucleus by three online programs, similar to that in poplar [11]. According to the results of NES prediction, there were 22 *PmC3H* proteins that might have the ability to shuttle between the cytoplasm and the nucleus to participate in signal transduction events. However, this speculation deserves further exploration.

Expression levels in different tissues showed that RR-TZF genes are expressed in multiple tissues, particularly in needles, suggesting that they may be involved in more common processes. In order to better understand the function of CCCH genes in abiotic response, we examined the expression patterns of five RR-TZF genes in two-year-old *P. massoniana* seedlings treated with multiple environmental stimuli. The RT-qPCR results revealed that all *PmRR-TZF* genes can respond to multiple stresses. For example, *PmC3H20* was significantly induced by MeJA and drought, which were performed 3.7 times and 5.2 times more than the control, respectively. *PmC3H32* and *PmC3H44* were significantly induced by PEG. The highest relative expression of *PmC3H40* appeared under ETH treatment, 4.71 times more than the control. the expression of *PmC3H45* peaked at 3 h under MeJA treatment, 9.14 times more than the control. Interestingly, the expression of all genes inhibited by ABA increased under serious drought treatment. However, RT-qPCR analysis showed that 12 CCCH genes in maize and *OsC3H10* were significantly induced by ABA and drought [12,35]. Therefore, it is necessary to investigate the ways in which RR-TZF genes increase the drought tolerance of *P. massoniana* in later experiments.

CCCH zinc finger proteins are important transcriptional regulators. In this experiment, we verified the transcriptional activation of five RR-TZF genes in yeast. Previous studies have shown that the conserved region of CCCH zinc finger proteins can bind to DNA, indicating that zinc finger proteins could regulate downstream genes when the upstream signal is received [36]. In addition, a typical conserved EELR domain in the C terminus of the *OsLIC* protein was confirmed to be vital for transcriptional activation. We also found the EELR domain in *PmC3H33* (Figure 3b). Whether it has the same function in *P. massoniana* needs to be further verified. In plants, the function of CCCH TZF proteins is still unclear. The function of plant CCCH zinc finger proteins was confirmed mostly in model plants such as *Arabidopsis* and *O. sativa*, but it is rare in trees, which are not sensitive to abiotic stress in a short period. This study provides the first systematic analysis of *P. massoniana* CCCH proteins, which have not been reported across the Pinaceae species. This analysis could not only contribute to select appropriate candidate genes for further functional study, but also be conducive in understanding the signal transduction and molecular mechanism of CCCH zinc finger proteins in plant stress resistance.

## 5. Conclusions

In this study, we identified 46 *PmC3H* genes from three transcriptomes of *P. massoniana* and performed bioinformatics analysis. These genes were divided into eight groups in total, and each gene contained at least one CCCH domain and several conserved motifs. Among the 46 genes, there were six TZF proteins which may be important in responding to different abiotic stresses. RT-PCR showed the expression levels of five RR-TZF genes under nine stress treatments in different time periods. Some proteins can be significantly expressed under specific stresses. The expression of selected genes all increased under serious drought stress. The expression of the partial CCCH family indicated that some CCCH genes play crucial roles in plant abiotic stress, especially under drought. A transcriptional activation activity assay showed five TZF proteins with transcriptional activation. At present, there are very few studies on CCCH transcription factors in forests, and this study first identified the CCCH family and TZF proteins in Pinaceae. This provides a theoretical basis for functional studies of the CCCH gene family and the select five candidate genes, *PmC3H20*, *PmC3H32*, *PmC3H40*, *PmC3H44 and PmC3H45*, which may play vital roles in responding to abiotic stress. In addition, these *PmC3H* genes demonstrate great potential for studying the stress resistance in *P. massoniana*.

## Figures and Tables

**Figure 1 genes-13-01639-f001:**
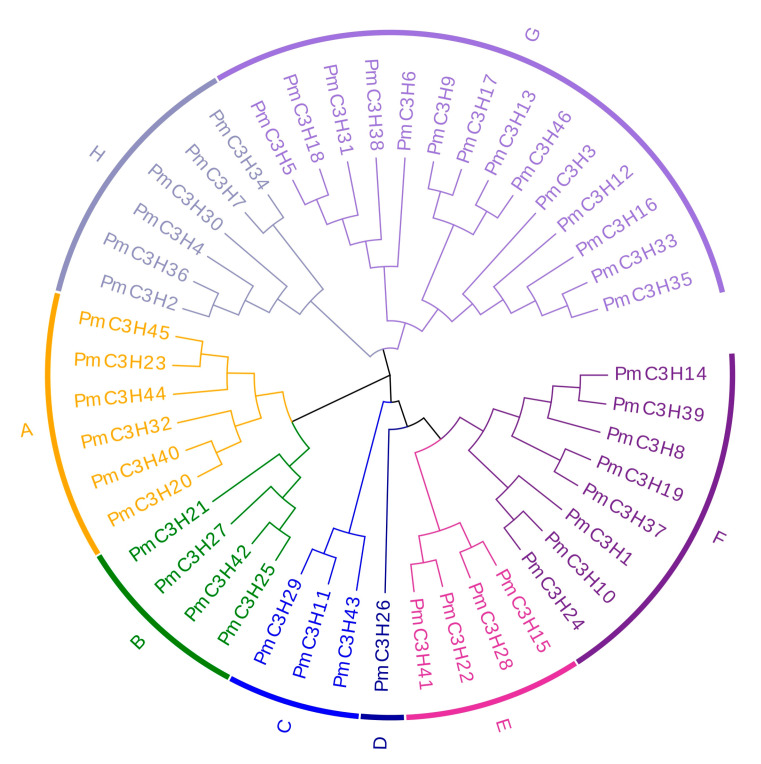
Phylogenetic analysis of CCCH proteins in *P**. massoniana*. Gene names in different colors correspond to groups of the same color in the outermost circle. The group is named from “A” to” H”.

**Figure 2 genes-13-01639-f002:**
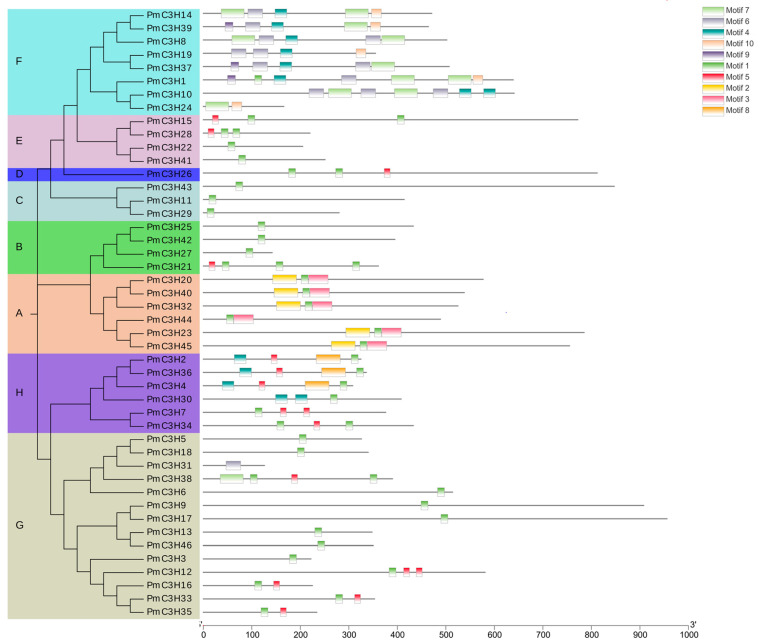
Phylogenetic analysis and motif distribution of 46 *PmC3H* proteins. Ten motifs present with ten different colors based on MEME analysis. The group is named from “A” to” H”.

**Figure 3 genes-13-01639-f003:**
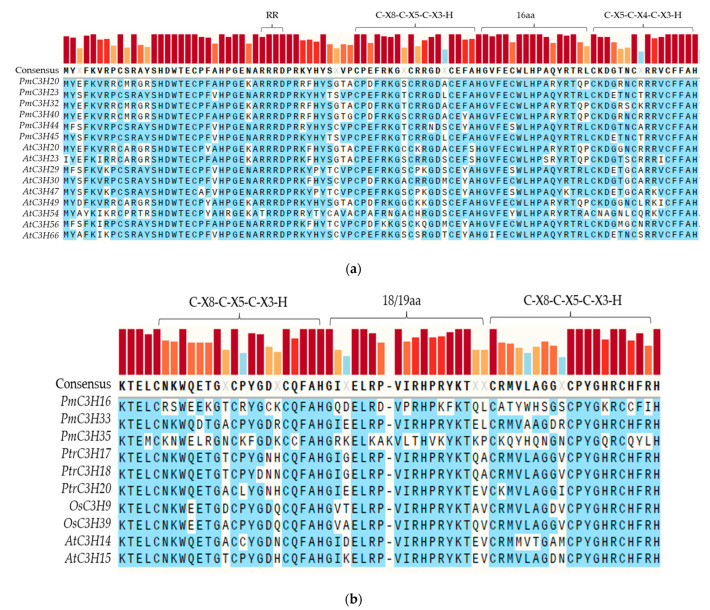
Multiple sequence alignment of different TZF proteins. (**a**) RR-TZF proteins in *P. massoniana* and *A. thaliana*, (**b**) typical tandem zinc finger (TZF) domain in *P. massoniana*, *P. trichocarpa*, *O. sativa* and *A. thaliana*.

**Figure 4 genes-13-01639-f004:**
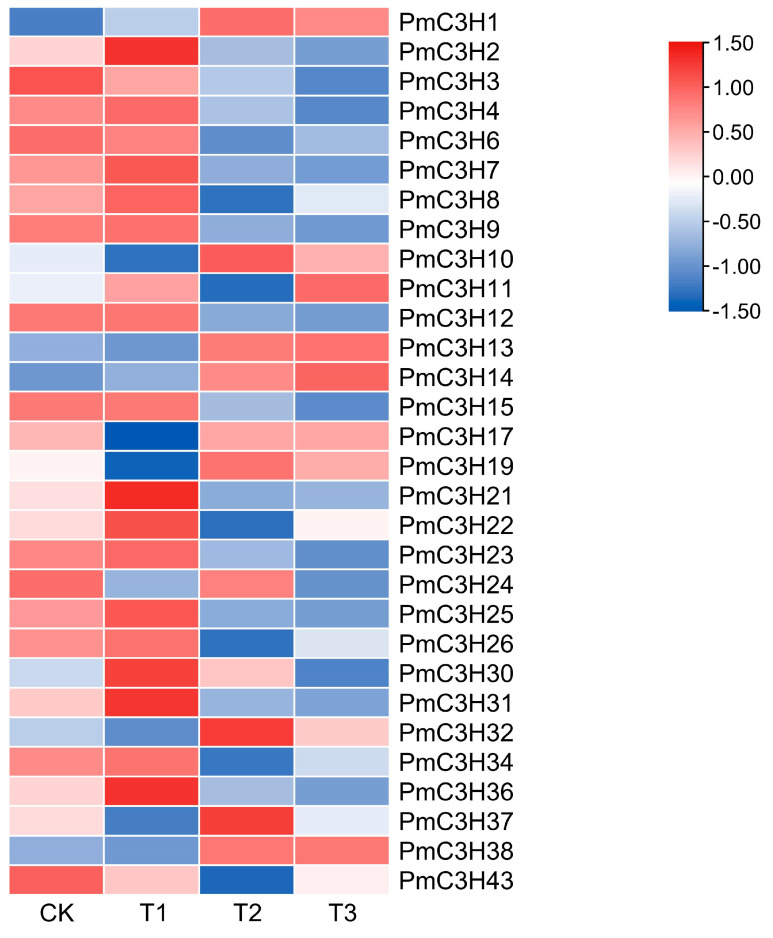
Transcriptional profiles of CCCH family members in *P. massoniana* under different drought levels: CK (80 ± 5)%, T1 (65 ± 5)%, T2 (50 ± 5)% and T3 (35 ± 5)%. Heat maps were generated using log2 (FPKM + 0.01) values, then performed row scale. The color scale represents relative expression levels based on the values of log2 fold change scale. Dark blue indicates a low expression level, light color indicates a medium level and red indicates a high level.

**Figure 5 genes-13-01639-f005:**
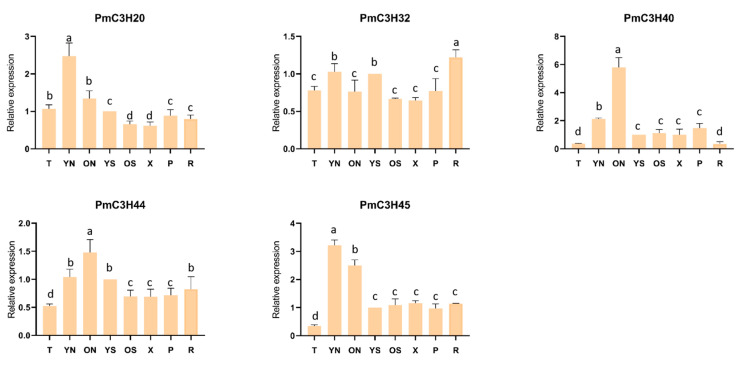
Relative expression of RR-TZF genes in eight representative tissues. T: Shoot apices; YN: young needle; ON: old needle; YS: young stem; OS: old stem; X: xylem; P: phloem; R: root. Same lowercase letter between different columns indicates no significant difference. Completely different lowercase letters between different columns indicate a significant difference, *p* < 0.05. The relative expression in YS was set as “1”.

**Figure 6 genes-13-01639-f006:**
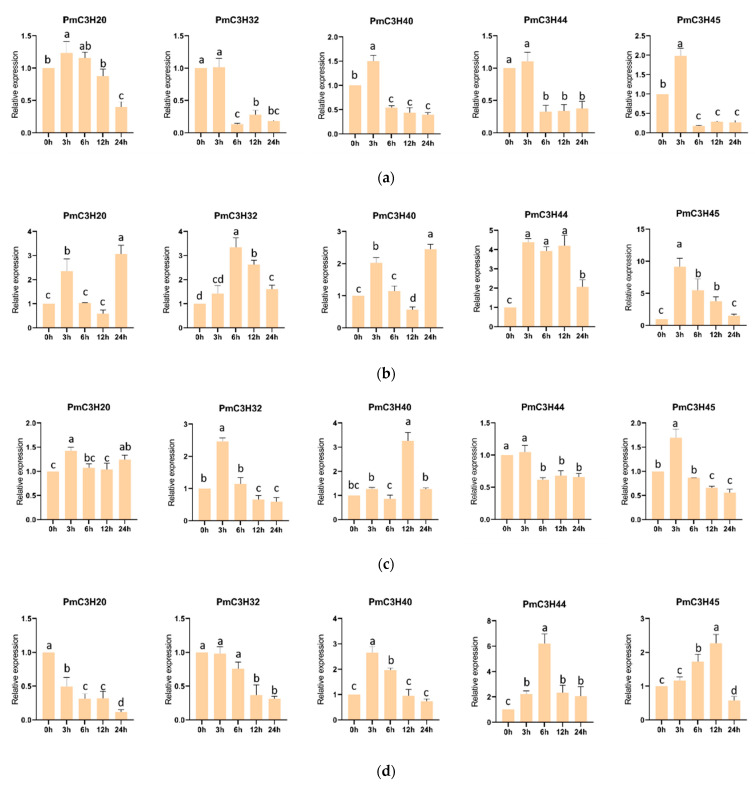
Expression profiles of five RR-TZF genes under different abiotic treatment. (**a**) ABA, (**b**) MeJA, (**c**) Mechanical injury, (**d**) SA, (**e**) H_2_O_2_, (**f**) ETH, (**g**) NaCl, (**h**) PEG, (**i**) Drought. Same lowercase letters between different columns indicate no significant difference. Completely different lowercase letters between different columns indicate a significant difference, *p* < 0.05. More than one lowercase letter in the same column indicate no significant difference between the column and other columns which contain one of the lowercase letters. The relative expression of 0 h was set as “1”.

**Figure 7 genes-13-01639-f007:**
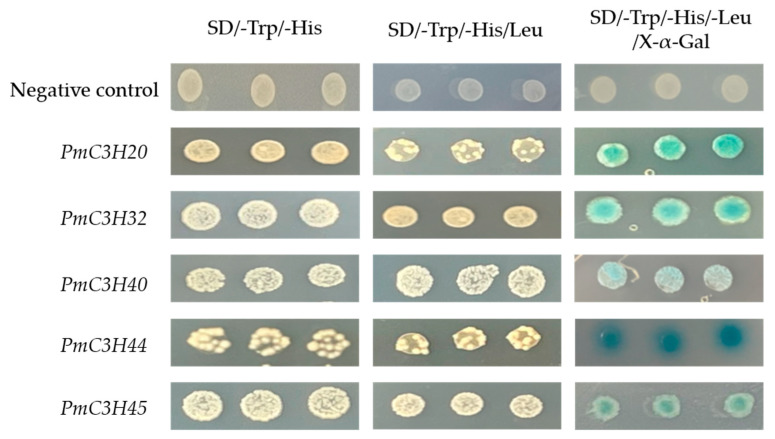
Transcriptional activation assay of five RR-TZF genes. Empty pGBKT7 vector was used as a negative control.

## Data Availability

The data presented in this study are available in Appendix A.

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
