# Peer review of "Transcriptome-Wide Identification of CCCH-Type Zinc Finger Proteins Family in Pinus massoniana and RR-TZF Proteins in Stress Response"

_genes, 2022, doi:10.3390/genes13091639_

Round 1

Reviewer 1 Report

The authors identified 46 CCCH zinc finger proteins in Pinus massoniana, and performed phylogenetic sequence analysis and protein domain/motif analysis for these C3H proteins. Focusing on a subgroup of these proteins, the authors examined transcriptional changes upon exposure to various abiotic stress conditions. Furthermore, they demonstrated that these C3H protein could activate transcription in yeast, suggesting transcriptional activity of C3H proteins might play a role during stress response in P. massoniana.

Overall, the findings are of interest and could be informative for future study, and the conclusions were supported by their analyses. However, I do think the manuscript could be improved. I have the following specific comments:

1. Figure 4 is confusing to me. If I understand correctly, the “CK” column represents log2FPKM while “T1-3” represents log2 fold change compared to CK. It would be confusing to plot them side by side without proper labels of which values were used. In addition, Line 264 “significantly changed”, what is the statistical significance of expression change for these genes? Providing statistical significance values for these genes in the heatmap or texts would address this concern.

2.Labels for statistical testing in Figures 6&7 is confusing, and it is not clear to me, what comparisons were done and what statistical results were, even referring to the figure legends. Along the same line, no indication of statistical method used in the method or figure legends.

3.At line 276, the statement “indicating that PmC3H genes have specific functions in resistance to drought stress” needs to be softened, as no direct support presented in the manuscript for this conclusion.

4.Figures 5-7 are not in proper numerical order, and this error should be fixed.

5.I think line 199 should be group “G” instead of group “D”.

6.No proper indications of subfamily “IX” or “I” in Figures S1 and S2.

7.Duplicating sentence in line 402.

8.The authors should proofread and correct grammar/spelling errors, i.e line 33 “Response”, several “expressed”, line 49 “mamma”.

Reviewer 2 Report

 Comments for the authors

Using transcriptomic data combined with bioinformatics approaches, authors have predicted CCCH-type proteins in Pinus massoniana. The authors determined the expression level of five of the predicted CCCH zinc finger proteins the different tissue and in various abiotic stresses. Further, using a yeast system, the authors have demonstrated the transactivation function of selected CCCH zinc proteins. The overall manuscript is well written and will provide a list of possible CCCH-zinc proteins found in P. massoniana.

Line 181- what does it mean that 5 ul of the diluted bacterial solution was collected?

Line 198-199, the largest group should be G and not D.

Figure 2 quality needs to improve; it is hardly readable.

The authors have not mentioned the statics method used in qRT-PCR analysis to determine the significance level. Moreover, it will be better if authors use a p-value and represent a star (*) to demonstrate the significance.      

Results from figure 7 can be represented in a better form, it’s too complicated to remember for the reader. Maybe the concise summary of findings can be represented in the form of a table. 

In figure 5, the transactivation function of the PmC3H40 is not clear. The authors should represent the quantification of the transcriptional activation strength of each RR-TZF gene. 

Most of the parts of the discussion section are repetitive to the results section, authors should make it more concise.

Line 402, repetitive sentence, "some proteins can be significantly expressed under specific stress"

Minor point- It is odd that both the text results and figure for Figure 5 are represented in the last paragraph of the result section, however, it is numbered abruptly as 5 out of the total 7 figures.  Figure 5 can be numbered as 7. Figures 6&7 can be numbered as 5 &6 respectively.
